# Machine Learning-Based Prediction of Well Logs Guided by Rock Physics and Its Interpretation

**DOI:** 10.3390/s25030836

**Published:** 2025-01-30

**Authors:** Ji Zhang, Guiping Liu, Zhen Wei, Shengge Li, Yeheya Zayier, Yuanfeng Cheng

**Affiliations:** School of Geology and Mining Engineering, Xinjiang University, Urumqi 830046, China; 18799632186@163.com (J.Z.); ping0991@163.com (G.L.); weizhen@xju.edu.cn (Z.W.); shenggeli0428@163.com (S.L.); yeheya0926@163.com (Y.Z.)

**Keywords:** machine learning, well logs, rock physics, SHapley Additive exPlanations (SHAP), explainable artificial intelligence (XAI)

## Abstract

The refinement of acquired well logs has traditionally relied on predefined rock physics models, albeit with their inherent limitations and assumptions. As an alternative, effective yet less explicit machine learning (ML) techniques have emerged. The integration of these two methodologies presents a promising new avenue. In our study, we used four ML algorithms: Random Forests (RF), Gradient Boosting Decision Trees (GBDT), Multilayer Perceptrons (MLP), and Linear Regression (LR), to predict porosity and clay volume fraction from well logs. Throughout the entire workflow, from feature engineering to outcome interpretation, our predictions are guided by rock physics principles, particularly the Gardner relations and the Larionov relations. Remarkably, while the predictions themselves are satisfactory, SHapley Additive exPlanations (SHAP) analysis uncovers consistent patterns across the four algorithms, irrespective of their distinct underlying structures. By juxtaposing the SHAP explanations with rock physics concepts, we discover that all four algorithms align closely with rock physics principles, adhering to its cause–effect relationships. Nonetheless, even after intentionally excluding crucial controlling input features that would inherently compromise prediction accuracy, all four ML algorithms and the SHAP analysis continue to operate, albeit in a manner that seems irrational and starkly contradicts the fundamental principles of rock physics. This integration strategy facilitates a transition from solely mathematical explanations to a more philosophical interpretation of ML-based predictions, effectively dismantling the traditional black box nature of these ML models.

## 1. Introduction

Well logs are capable of capturing a diverse array of geophysical parameters within the penetrated formations, encompassing caliper measurements, acoustic velocity, gamma-ray radioactivity, neutron density, spontaneous potential, resistivity, nuclear magnetic resonance, and more. These vast well logging datasets are extensively employed in comprehensively analyzing the physical attributes of the target strata, offering indispensable insights into the geological composition and potential of hydrocarbon reservoirs [1,2].

Rock physics models are typically employed in the refinement of initially acquired well logs, aimed at correcting inaccuracies or predicting missing data segments [3,4]. These models bridge the gap between macroscopic measured seismic properties and microscopic rock features, such as porosity, resistivity, mineralogy, fluid saturation, and pore pressure [5,6,7].These models can be broadly categorized into two types: empirical and physically based [8]. Physically grounded models, such as Gassmann’s equation, often hinge on certain assumptions, thus restricting their scope of application [9,10,11,12]. In contrast, empirical models, exemplified here by the Gardner relations and the Larionov relations, are founded on experimental data and stem from straightforward regression analyses [13,14,15,16,17]. Despite their simplicity, these empirical models have demonstrated remarkable practicality and widespread applicability in the petroleum industry [18,19].

Alternatively, machine learning (ML) algorithms offer a solution for correcting errors or filling in gaps in well logs [20,21]. In recent years, the application of ML technology has experienced a dramatic surge in popularity and adoption across the logging industry, marking a significant leap in technological progress [22,23,24,25]. While rock physics relies on explicit knowledge gained through extensive experience and experimentation, ML algorithms depend on data training [26,27]. This often renders ML a black box with limited interpretability, significantly reducing its perceived enhanced value and credibility within the industry [28,29,30,31]. Explainable artificial intelligence (XAI) serves as an umbrella term, encompassing the incorporation of human expertise into ML workflows [32,33,34,35]. This integration spans a wide spectrum, from precisely defined equations (PINN) where human knowledge is explicit [36,37,38], to more implicit and less readily articulated experience that individuals possess [39,40]

In this study, we propose to integrate the SHapley Additive exPlanations (SHAP) framework and rock physics into the workflow for predicting well logs, leveraging four distinct ML algorithms. This approach strives to dismantle the traditional opacity of ML models, thereby fostering unprecedented transparency. Specifically, SHAP allows us to quantify, on a localized level, the precise impact of each input feature on individual predictions, while also presenting a holistic, global overview across the entire dataset [41,42,43]. Furthermore, rock physics serves as a pivotal guide, quality guardian, and reference point throughout the ML-based workflow, spanning from feature engineering to results interpretation. By leveraging this integration, we aim to not only bolster the accuracy but also elevate the interpretability of ML-based well log predictions, thus propelling the field forward towards more reliable and intelligible outcomes.

## 2. Methodology

### 2.1. Rock Physics

To describe the dependency between bulk density and compressional velocity, a set of relations were formulated as [16](1)$$\rho_{b}=a{V_{p}}^{b}$$
where *ρ*_b_ is bulk density in g cm^−3^, *V*_p_ is the P-wave velocity in ft s^−1^, and a and b are lithology-specific constant coefficients that vary across different rock types.

In this study, the Gardner relations were initially used to calibrate the correlation between acoustic logs and density logs. Subsequently, these relations were used to generate density logs in cases of suspicion or inaccuracies. It is noteworthy that, besides acoustic data, density can also correlate with shear wave velocity, neutron porosity, or gamma ray [44]. Therefore, in addition to the Gardner relations, a machine learning-driven approach can be implemented to generate density logs from a diverse range of inputs.

To describe the clay volume fraction in a rock, the gamma index, *I*_c_, was introduced [14], which is simply a normalized representation of the recorded nature gamma logs. Specifically,(2)$$I_{c}=\frac{GR-{GR}_{min}}{{GR}_{max}-{GR}_{min}}$$
where *GR* is the measured value of gamma logs expressed in American Petroleum Institute (API) units, and *GR*_max_ and *GR*_min_ are the respective maximum and minimum values encountered in shales and sands, with these thresholds herein specified as 140 and 20 API units.

Since the gamma index tends to exceed the actual clay volume fraction, *V*_c_, alternative modified equations are often employed [14]:(3)$$V_{c}=0.083(2^{{3.7I}_{c}}-1)$$

The formula holds applicability for both the Tertiary period and more recent geological epochs. The gamma logs capture the natural gamma radiation emanating from radioactive elements such potassium (K), uranium (U), and thorium (Th), which are primarily present in clays. However, it is worth noting that non-shale radioactive mineral like Sylvite, feldspars, and micas can also contribute to gamma logs, potentially introducing inaccuracies [45,46,47,48]. Therefore, for specific geological problems, the above equations may be further modified or machine learning algorithms can be employed, using either gamma logs alone or a combination of multivariate inputs including gamma logs, density, P-wave velocity, and acoustic impedance, among others.

### 2.2. Data

Nestled in the southern North Sea basin, the F3 block (54.8669° N, 4.8131° E) in the Dutch sector lies atop the Central Graben, bordered by the Ringkøbing-Fyn High to the east and the Mid-North Sea High to the west. Geologically, the Mid-Miocene Unconformity (MMU) serves as a demarcation line, separating the Cenozoic into the underlying Paleogene units and the overlying Neogene units [49]. The Paleogene units beneath have been structurally influenced by the syn-depositional halokinetic movements of the underlying Zechstein evaporates, manifesting in the formation of faults and drapes. In contrast, the Neogene units overlaying them consist primarily of coarse-grained prograding sediments with an overall high porosity of 22–30%. These sediments have formed immense polycyclic fluvio-deltaic systems, a direct response to high-frequency relative sea-level cycles [50]. Evidence of these cycles can be seen in large-scale sigmoidal beddings of truncations, with onlaps onto them to the east, and downlaps to the west [51,52]. Within this geological context, our study focuses specifically on a select interval of log data (FS2–FS6) obtained from four wells within the F3 block, providing insights into a subset of these remarkable fluvio-deltaic systems.

The 3D seismic data and log data for this study originate from the F3 block, with the seismic data acquired in 1987 by NAM for petroleum exploration purposes [53]. dGB Earth Sciences has kindly provided the upper segment of the seismic data, encompassing a time range of 0 to 1.848 s. The inline range ranges from 100 to 750, while the crossline range extends from 300 to 1250, both with a 25 m line spacing, covering a total area of 386.93 km^2^. An inline is shown in Figure 1b. Four vertical wells inside the survey are provided: F02-1, F06-1, F03-2, and F03-4, all measured by NLOG. These wells are equipped with sonic and gamma ray logs. However, it is noteworthy that according to dGB’s records, only F02-1 and F03-2 have density logs available. For F03-4 and F06-1, density logs were predicted using a neural network model trained on data from the first two wells, using sonic and gamma logs as input. Additionally, the original porosity logs for all wells were calculated from density data based on a linear formula [3,13]:(4)$$\phi =\frac{\rho_{ma}-\rho_{b}}{\rho_{ma}-\rho_{fl}}$$
where *ϕ* is porosity, and *ρ*_ma_ and *ρ*_fl_ are the densities of the matrix and mud filtrate respectively, with *ρ*_ma_ set to 2.65 g cm^−3^ and *ρ*_fl_ set to 1.05 g cm^−3^. Furthermore, for this study, the clay volume fraction was calculated from gamma logs using the Larionov relations.

Figure 2 gives a comprehensive overview of the seven logs corresponding to the four distinct wells. For each well, a specific interval ranging from FS2 to FS6 is selected and presented. In comparison to F03-2 and F03-4, the depths recorded for F02-1 and F06-1 are notably deeper and shorter. This difference is due to their geographic position, which lies in the southwestern periphery of the survey area, precisely at the foreland of the prograding delta, as depicted in Figure 1.

### 2.3. Machine Learning

Four different popular ML algorithms were used, comprising Linear Regression (LR), Multi-Layer Perceptron (MLP), Random Forests (RF), and Gradient Boosting Decision Tree (GBDT). LR, without any regularization, as the first type of regression analysis to undergo rigorous study and extensive practical application, is widely used owing to its simplicity, interpretability, and ease of implementation [54]. MLP, a commonly applied artificial neural network model, boasts robust expressive and generalization capabilities, enabling it to tackle non-linear problems and high-dimensional data [55,56,57]. Both GBDT and RF are ensemble learning techniques rooted in decision trees. GBDT sequentially constructs multiple decision trees, with each subsequent tree aiming to correct the prediction errors of its predecessor, optimizing predictive performance by gradually minimizing residuals [58]. Conversely, RF concurrently builds numerous decision trees, utilizing averaging for regression and voting for classification, thereby mitigating the risk of overfitting and often yielding more stable and accurate predictions [59].

Due to the limited availability of only four well logs, we opted to utilize three of them for training the ML algorithms, reserving the remaining well for validating the predictions of the trained models. However, in scenarios where there were suspicions of inaccuracies in these logs, we relied on rock physics principles to calibrate and reconstruct the logs as replacements. As detailed in the Data section, we were able to generate density logs employing the Gardner relations, while *V*_c_ logs were derived utilizing the Larionov relations.

Standard scaling was applied to the input features for MLP and LR, whereas it was omitted for the two tree-based algorithms. The coefficient of determination (R^2^) and root mean squared error (RMSE) were used as metrics to evaluate the accuracy of predictions made by ML algorithms. Furthermore, to interpret the mathematical principles of how these ML algorithms operate, both locally and globally, we used the SHapley Additive exPlanations (SHAP) framework [41]. Based on the solid theoretical foundation of cooperative game theory and the utilization of Shapley values, SHAP offers quantitative insights into the specific contributions of each feature to individual predictions, as well as their collective influence across the entire dataset [60,61]. This approach not only facilitates the assessment of ML algorithm performance but also serves as a valuable guide for feature selection and model design optimization.

The Python scripts used in this study for ML and SHAP analysis were based on the Scikit-learn and SHAP libraries [62,63].

Our primary objective is to elucidate the mathematical frameworks that underpin ML algorithms and interpret the causality behind their predictions within the context of rock physics. As such, we prefer to use standard and straightforward ML methods to facilitate their explanations and to derive conclusions that are broadly applicable. In interpreting SHAP, it is crucial to concentrate solely on the influence of input features by maintaining the consistency of the ML models. Our benchmarks, detailed in Appendix A, reveal that optimizing the models’ hyperparameters did not significantly improve their performance compared to using fixed settings (Table A1 and Table A2), but it did lead to notable changes in SHAP explanations (Figure A1). Therefore, we consistently avoid making any modifications based on optimization outcomes.

To address variability from random initialization, we tested 40 random seeds and found consistent results within each stochastic model, except for the MLP which showed significant fluctuations. This is shown in Figure A2.

## 3. Results and Discussions

### 3.1. Porosity

First, we show the results of ML-based prediction of porosity logs.

Figure 3 shows the density logs of the four wells, before and after correction using the Gardner relations. The original density of F02-1 is significantly lower than the other three wells. Since density logs were calculated from measured acoustic logs, we used the Gardner relations to fit the density of the three wells, where in the exponential function we optimized the coefficient to be 0.1018 and the index to be 0.3445. Relying on the Gardner relations, density logs of F02-1 were corrected using acoustic logs as inputs.

To gain a deeper understanding, we compiled a crossplot that illustrates the interrelationships among the seven log data sets, as shown in Figure 4. Notably, we have incorporated an additional well, F02-1Cor, which is a corrected version of F02-1. In this updated version, the density was fine-tuned using the Gardner relations, leading to a subsequent recalculation of the porosity. As observed in the crossplot comparing *ρ*_b_ and *AC*, the data points of F02-1 stand out conspicuously, indicating the need for further scrutiny. The corrected F02-1Cor, however, exhibits a seamless alignment with the other wells, demonstrating the effectiveness of the corrections. Moreover, we recalculated *V*_p_ as the reciprocal of *AC*, and consequently, *AI* was recalculated as the product of *ρ*_b_ and *V*_p_. This corrected well, and F02-1Cor, exhibits an exceptional correlation across all seven logs, testament to the rigor and reliability of our findings. However, as Figure 4 indicates, the correlations are poor between gamma logs (and consequently *V*_c_) and elastic logs, including *AC*, *V*_p_, *ρ*_b_, *AI*, and *ϕ*. This incongruence in characteristics could potentially influence ML-based predictions, and its implications will be discussed in the next section.

Figure 5 presents a comparative analysis of the ML-based prediction of porosity logs, highlighting the difference between the pre- and post-correction states of density logs. In this process, four parallel ML algorithms were used, specifically RF, GBDT, MLP, and LR. The training of these ML algorithms encompassed data from three wells, whereas the remaining well served as the testing ground. Notably, the problematic density logs from F02-1 posed a challenge.

In the pre-correction scenarios, two distinct approaches were adopted for ML training and testing: either including F02-1 in the training set or reserving it for testing. Given that porosity measurements were not readily available for all wells and were instead estimated from density logs using a linear formula, density itself was excluded as an input variable for ML training. Similarly, *V*_p_ and *AI* were also excluded, leaving only the measured logs, comprising *AC* and *GR*, as the input features.

Remarkably, for all four ML algorithms, the prediction accuracy of pre-correction states was deemed unsatisfactory. However, an interesting trend emerged when the problematic F02-1 was included in the training dataset, resulting in improved prediction performance compared to when it served as the testing well. Furthermore, the prediction accuracy of all four algorithms underwent a significant enhancement when both acoustic and gamma logs were used as inputs, surpassing the accuracy achieved by relying solely on acoustic logs. Despite the fact that gamma logs display a notably feeble correlation with porosity, as depicted in Figure 5, their integration with acoustic logs contributes to a more precise prediction.

After the correction of density logs, however, the prediction accuracy for all four ML algorithms experienced a substantial surge, achieving exceptionally high levels of accuracy, with R^2^ values approaching 1 for each algorithm. This trend was observed when F03-4 served as the testing well, and the corrected F02-1 was incorporated into the training dataset, as shown in Figure 5c. It is crucial to emphasize that the achieved optimal performance hinges critically on the establishment of the ideal dataset setting, where the porosity data is systematically derived from the measured acoustic data through the application of a distinct and carefully calibrated formula. This underscores the paramount importance of a rigorous and systematic approach to data processing and model training, which is essential for attaining predictions of both high fidelity and precision.

To gain a deeper insight into the mechanisms underlying ML-based predictions, global SHAP values were calculated to interpret the results. These analyses consistently pointed to acoustic logs as having an overarching influence on predicting porosity, regardless of the ML algorithm, training and testing scenarios, or prediction accuracy, as summarized in Figure 5d. Conversely, the other input logs, namely gamma logs, were largely disregarded, with the exception of one instance when the problematic F02-1 log was incorporated as an input. In this particular scenario, the inclusion of gamma logs resulted in a marginal improvement in prediction accuracy. This interpretation aligns with the weak correlation exhibited between gamma logs and porosity logs, as illustrated in Figure 4. Generally, MLP exhibits a greater reliance on gamma logs compared to the other three algorithms.

### 3.2. Clay Volume Fraction

Here we present the outcomes of ML-based prediction of clay volume fraction (*V*_c_) logs and their interpretations.

Typically, *V*_c_ is derived from gamma logs, which records the natural gamma radiation emitted from radioactive elements present in clays. In this study, the Larionov relations were used to calculate *V*_c_ from the measured gamma logs. As depicted in Figure 4, *V*_c_ theoretically exhibits a strong correlation with gamma logs, but it correlates weakly with acoustic logs and other logs derived from acoustic logs, such as density, acoustic impedance, P-wave velocity, and porosity. Figure 6c shows the prediction of *V*_c_ for F03-4 when the inputs for training the ML algorithms included logs that correlate poorly with *V*_c_. Specifically, the ML algorithm used was RF. The training data originated from the other three wells, spanning the interval between FS2 and FS4. As anticipated, the prediction performance was unsatisfactory. However, when gamma logs were incorporated into the training process, the prediction accuracy was improved significantly, as shown in Figure 6c. Similar results were observed when F03-2 served as the testing well and the other three wells as the training well, as shown in Figure 6e. Nonetheless, there were sections of the *V*_c_ logs for both F03-4 and F03-2 that remained inaccurately predicted.

To gain a more comprehensive understanding of the prediction’s performance using RF and the other three ML algorithms, we calculated both global and local SHAP values for each individual prediction. In the case of predicting *V*_c_ for F03-4 using RF, gamma logs emerged as the predominant input feature, as explained by both the SHAP summary plot and decision plot (comparing Figure 6b with Figure 6d). This finding aligns with the anticipation based on human knowledge, as gamma logs are the physical trigger for the effect, namely *V*_c_. Meanwhile, the impact of the other four input features is minimal. However, in scenarios where gamma logs are absent from the training dataset, the accuracy of *V*_c_ predictions declines significantly, rendering the importance of the four remaining input features indispensable. In such a situation, irrespective of the prediction inaccuracy, and regardless of the weak correlation between the input features and the prediction target, the ML algorithm relies heavily on the provided inputs, as it lacks alternative data sources. When assessing the impact on the inaccurate prediction results, the four input features are approximately of equal significance. This trend is observed across the other three algorithms, as shown in Figure 7b. However, if a ranking is necessary, acoustic logs emerge as the most crucial input except for LR, likely attributed to their status as the only measured feature among the inputs.

The interpretation of the significance of various features across all four ML algorithms is summarized in Figure 7. Notably, although MLP initially failed to prioritize gamma logs as the primary input feature, it regained its rationality after undergoing a noise-robust training procedure. This can be observed in Figure 7b for MLP, which demonstrates the algorithm’s adaptability. Within this noise-robust training paradigm, the predictions attained satisfactory levels, albeit with a notable decline in MLP’s performance due to a relatively low signal-to-noise ratio (SNR) of 30, as evident in Figure 7a for MLP. The SHAP analysis revealed consistent patterns among all four algorithms, regardless of their varied underlying architectures—whether tree-based, neutron network-driven, or linear regression-oriented. By comparing the SHAP explanations with rock physics principles, we observed a close alignment of the four algorithms with the fundamentals of rock physics, indicating adherence to its cause relationships.

To interpret the sections of the *V*_c_ logs that exhibited unpredictable patterns at a depth of approximately −960 m, a review of the training and the testing dataset was undertaken. Specifically, considering the scenario where F03-4 serves as the testing well and the other three wells as the training well, we note that the *V*_c_ range for the other three wells ranges from 0.050 to 0.300, whereas the *V*_c_ range for F03-4 to be predicted is 0.025 to 0.200. Therefore, the predicted lowest values of *V*_c_ for F03-4 exceed the bounds of the training dataset, which is indicative of an Independent and Identically Distributed (IID) issue. Analogously, reversing the roles of the testing and training well also leads to an inaccurate prediction of the higher *V*_c_ values in F03-2. The lowest and highest unpredictable *V*_c_ values are positioned at the periphery of both the SHAP summary plot and the decision plot, as indicated by the grey arrows in Figure 6c,d, and the yellow arrows in Figure 6e,f.

To further elaborate on the intricacies of the IID problem and explain the inability to make accurate predictions, we employed a straightforward binary logarithm function of *y* = log_2_*x*, with the range of *x* set from 1 to 11. We subjected the prediction of *y* to a rigorous test using four different ML algorithms, contrasting the results against those obtained from cubic spline interpolation and the ground truth. As shown in Figure 8a at *x* = 2.4, where the true value of *y* is denoted by black crosses along the black line, RF predicted *y* = 1.291, marked by a black dot, exceeding the actual value. From *x* = 2.0 to 2.8, the predictions either exceeded or fell below the true values, as evident in the lower inset. Notably, as we incrementally removed one or two neighboring labels from the training dataset, the predicted values rose significantly, depicted by blue dots. Conversely, the predicted values decreased progressively when one or two labels to the right were eliminated. A comparison with the ground truth is illustrated in the upper inset. Similar trends were observed for the *x* range from 9.0 to 9.8, as shown in Figure 8b. The performance of the remaining three algorithms is presented in Figure 8c–e. Despite the remarkable success ML has achieved owing to its numerous advantages, the predictions generated by the four ML algorithms can still trail behind the accuracy of cubic spline interpolation, even with a substantially vast training dataset.

These findings underscore the sensitivity of ML-based predictions to the training dataset. Furthermore, as neighboring samples are gradually removed, the IID assumption becomes invalid, leading to an out-of-distribution (OOD) problem that ultimately compromises the predictive accuracy.

## 4. Conclusions

We have integrated SHAP and rock physics into the ML-based approach for predicting well logs, specifically targeting porosity and clay volume fraction. While ML excels in handling complex, nonlinear challenges that involve diverse input features and extensive datasets, its operational mechanisms are often less mathematically transparent compared to traditional algorithms such as least squares and Bayesian methods. Conversely, rock physics, with its clear and intuitive foundations, faces constraints in its applicability due to the numerous inherent assumptions, hampering its capacity to address ambiguous or uncertain scenarios.

The union of these two methodologies has yielded significant advantages in our study. Rock physics has played a pivotal role in feature engineering, selection, algorithm comparison, and the comprehensive evaluation of four distinct algorithms: RF, GBDT, MLP, and LR. Despite their varied architectures—tree-based, neutron network-driven, or linear regression-oriented—the SHAP analysis has uncovered consistent operational patterns across these algorithms in noise-robust training and testing scenarios. Moreover, the SHAP explanations closely align with the causal relationships outlined in rock physics principles. However, when crucial controlling input features were intentionally omitted, affecting prediction accuracy, all four ML algorithms and the SHAP analysis continued to operate in a seemingly irrational manner, starkly contrasting with the fundamental principles of rock physics. By comparing SHAP explanations with rock physics concepts, we have established parallels and contrasts with human knowledge, inherently rooted in causal relationships. The invaluable guidance provided by rock physics has empowered us to intervene in ML-based predictions, not only enhancing our comprehension of the strengths and limitations of ML models but also potentially paving the way for optimizing the performance of these workflows. This approach has propelled a transition from purely mathematical explanations to a more philosophical interpretation of ML-based predictions, effectively dismantling the traditional black-box nature of ML models.

## Figures and Tables

**Figure 1 sensors-25-00836-f001:**
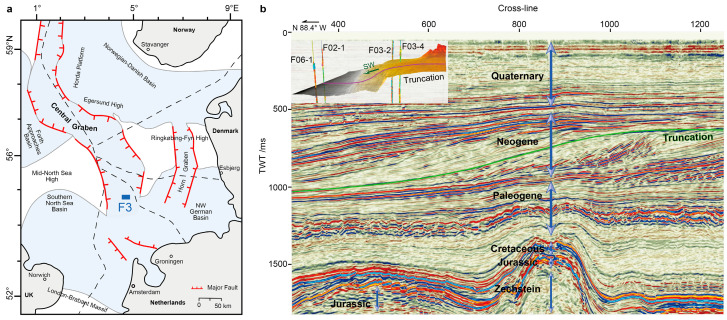
Geology setting of the F3 block. (**a**) Structural setting of the North Sea basin. The location of the F3 block is highlighted by a blue rectangle. (**b**) Seismic section of the F3 block (Inline 362). The geological periods of strata are annotated. The green line is the Truncation horizon. The inset is the Truncation horizon in 3D, with the four wells and the prograding direction annotated. The red line serves as the intersection between the horizon and the seismic section.

**Figure 2 sensors-25-00836-f002:**
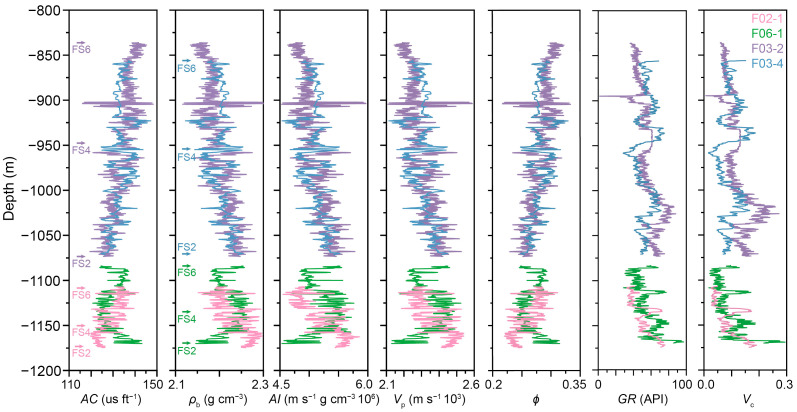
Log data obtained from the four wells. From **left** to **right**: acoustic, bulk density, acoustic impedance, P-wave velocity, porosity, gamma, clay volume. The top of each interval in each well is denoted using the same color as the line of the logs.

**Figure 3 sensors-25-00836-f003:**
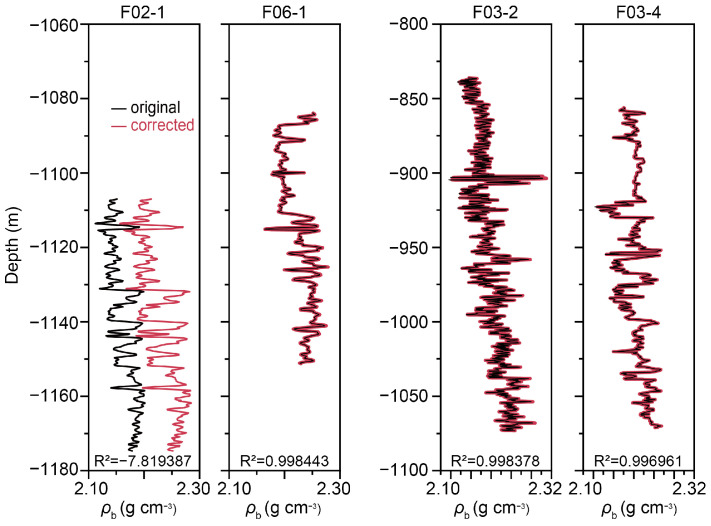
Density logs of the four wells before and after correction using the Gardner relations.

**Figure 4 sensors-25-00836-f004:**
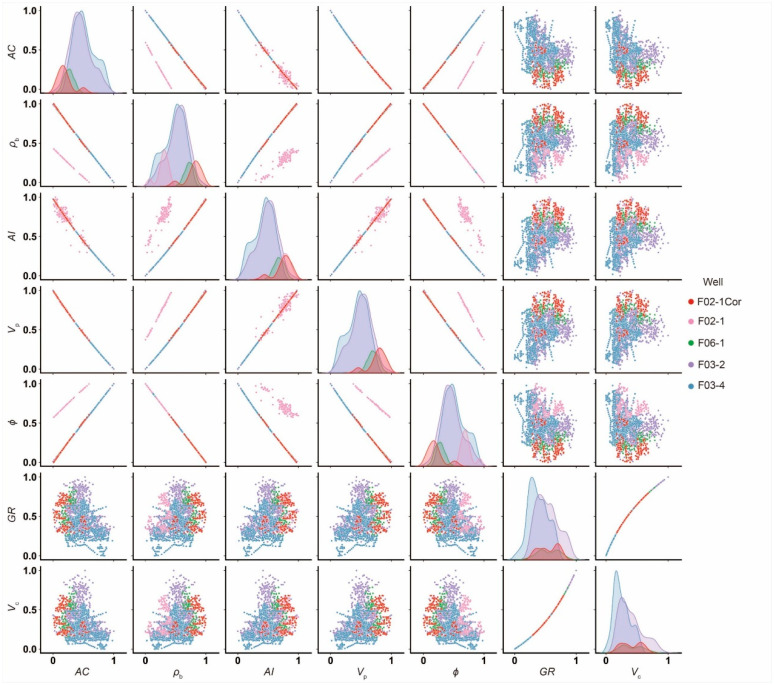
Crossplot of the seven logs of the four wells and F02-1Cor. From **left** to **right**, *AC*, *ρ*_b_, *AI*, *V*_p_, *ϕ*, *GR* and *V*_c_.

**Figure 5 sensors-25-00836-f005:**
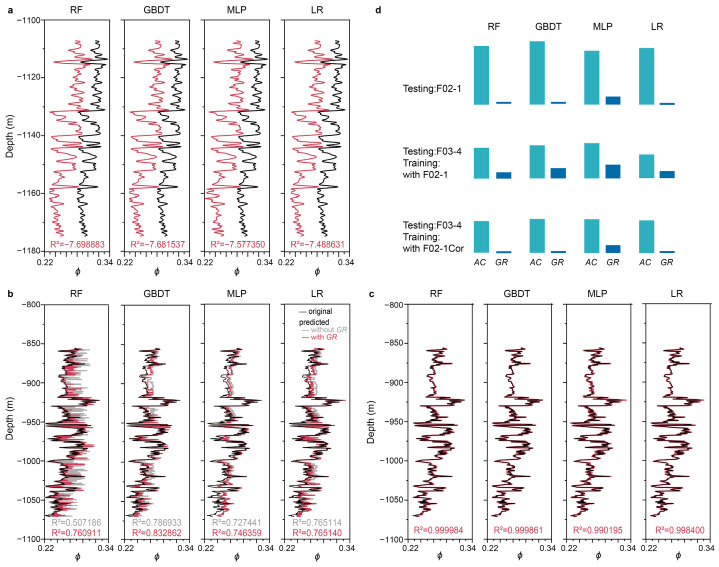
Prediction of porosity logs with original and corrected density logs using four ML algorithms. (**a**) F02-1 as the testing well. (**b**) F03-4 as the testing well. F02-1 was included in the training dataset. (**c**) F03-4 as the testing well. F02-1Cor was included in the training dataset. (**d**) SHAP explanation for these predictions. Top, for cases in both (**a**) and the cases where F02-1Cor served as the testing well. Middle, for cases in (**b**). Bottom, for cases in (**c**).

**Figure 6 sensors-25-00836-f006:**
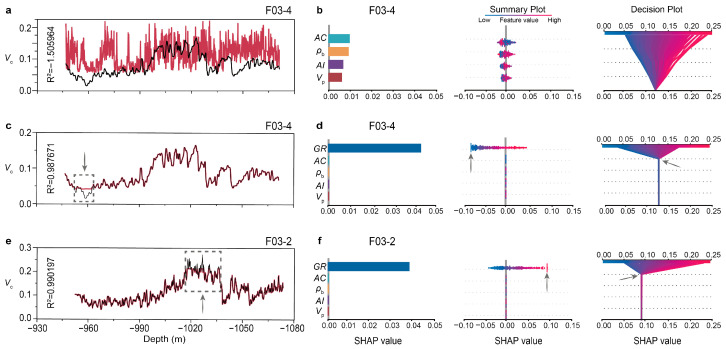
Prediction of clay volume fraction using RF and the SHAP explanations. (**a**) Prediction for F03-4 where gamma logs were excluded as inputs. (**b**) Feature importance for (**a**) as explained by SHAP summary plot and decision plot. (**c**) Prediction for F03-4 where gamma logs were incorporated as inputs. (**d**) SHAP explanations for (**c**). (**e**) Prediction for F03-2 where gamma logs were incorporated as inputs. (**f**) SHAP explanations for (**e**). Arrows indicate unpredictable samples.

**Figure 7 sensors-25-00836-f007:**
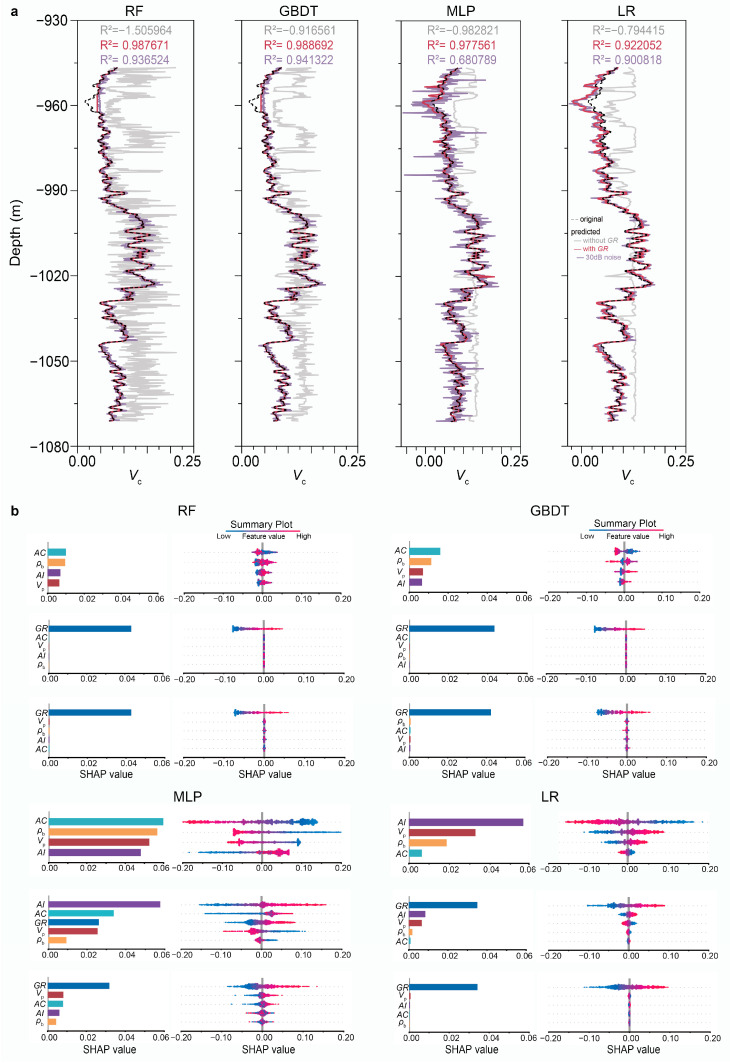
Accuracy and feature importance in predicting clay volume fraction for F03-4 as explained by SHAP summary plot for four ML algorithms. (**a**) Predicted *V*_c_ logs under three different scenarios. In contrast to a standard scenario, two variations were tested: one excluded GR in training, and the other employed a noise-robust training and testing approach with an SNR of 30. (**b**) Both local and global SHAP explanations for these three different prediction scenarios. For each algorithm: Top, training without GR. Middle, standard training and testing with GR. Bottom, noise-robust training and testing.

**Figure 8 sensors-25-00836-f008:**
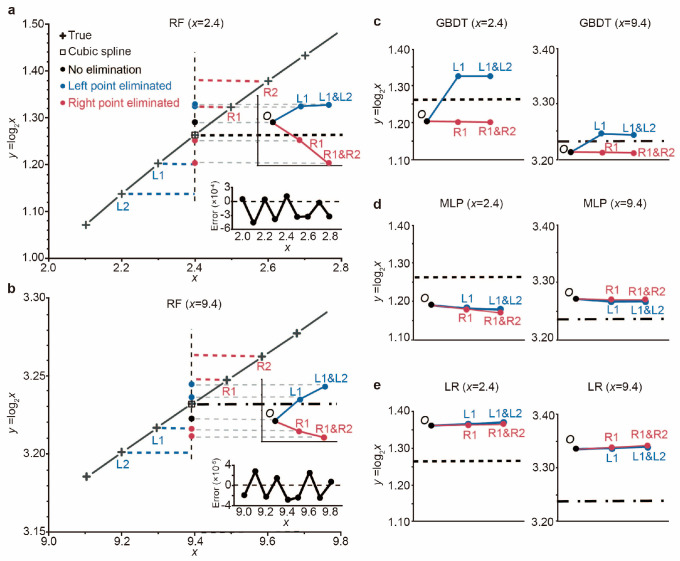
ML-based prediction of the binary logarithm function. (**a**) For *x* range from 2.0 to 2.8. Black line, theoretical line. Black crosses, theoretical values. Dots, ML-based predictions. Upper inset, predictions versus truth as neighboring samples eliminated from training. Lower inset, prediction errors at each testing point. (**b**) For *x* range from 9.0 to 9.8. (**c**) For GBDT. **Left**, *x* = 2.4. **Right**, *x* = 9.4. (**d**) For MLP. (**e**) For LR.

## Data Availability

The custom Python scripts and data associated with this research are available and can be accessed via the following URL: https://gitee.com/zhangjixju/ml-code/tree/master (accessed on 13 January 2025).

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
