# Peer review of "Machine Learning-Based Prediction of Well Logs Guided by Rock Physics and Its Interpretation"

_sensors, 2025, doi:10.3390/s25030836_

Round 1
Reviewer 1 Report
Comments and Suggestions for Authors
The paper “Machine learning-based prediction of well logs guided by rock physics and its interpretation” considers an important problem of machine learning applicability and reliability for well data log interpretations. The main conclusion of the paper is that the union of ML approach and rock physics will enhance the performance of both approaches. The paper is well written and designed. I have three small remarks:
1. The authors should have indicated which frameworks for ML and SHAP they used.
2. The authors used simple, well-known ML algorithms. They are used by everyone and for everything, and they have also been used more than once for borehole logs. I recommend the authors to present examples of such use by other geophysicists and compare the results with their own.
3. From a programming point of view, everything looks simple there, all the used methods are in the standard libraries.
Reviewer 2 Report
Comments and Suggestions for Authors
The manuscript addresses an important topic, integrating machine learning (ML) and rock physics for well parameter prediction. The combination of SHAP interpretability and rock physics principles has potential to advance the understanding of ML model behavior in geoscience. However, the current execution of the study requires significant methodological and reporting improvements to meet the standards of reproducible and reliable scientific research.
1) The results are based on single runs of stochastic models (MLP, RF, and GBDT). As these models rely on random initialization, the approach proposed by the authors do not account for variance introduced by random initialization. To address this, each model should be evaluated through at least 30 independent runs using different random seeds.
2) A critical limitation of this manuscript lies in its inadequate reporting of machine learning model specifications and hyperparameters, which impacts its reproducibility and scientific rigor. The authors employ four different models - Gradient Boosting Decision Trees (GBDT), Random Forest (RF), Multilayer Perceptron (MLP), and Linear Regression - yet fail to provide essential implementation details for any of them.
2a) for the tree-based models (GBDT and RF), the manuscript omits hyperparameters that significantly influence model behavior and performance. In the case of GBDT, there is no specification of the number of trees, learning rate, maximum tree depth, minimum samples per leaf, or feature sampling strategies. Similarly, for Random Forest, the authors do not report the number of trees in the ensemble, maximum depth settings, minimum sample split criteria, or the chosen bootstrap sampling strategy. These parameters are fundamental to understanding the models' complexity and their capacity to generalize.
2b) the neural network implementation (MLP) lacks even more critical details. The authors provide no information about the network architecture, including the number and size of hidden layers, choice of activation functions, or training specifications such as learning rate, optimizer selection, batch size, and number of epochs. The absence of information about regularization strategies (such as dropout) and early stopping criteria makes it impossible to assess how the authors addressed potential overfitting issues.
2c) even for the simpler Linear Regression model, important details are missing. The manuscript does not specify whether any regularization techniques were employed (L1 or L2) or what feature scaling methods, if any, were applied to the input data. While Linear Regression has fewer hyperparameters, these details remain crucial for reproducibility.
3) The manuscript does not evaluate the significance of performance differences between models. It is recommended to conduct an ANOVA to test for significant differences or include a post-hoc analyses such as Tukey test for pairwise comparisons.
4) Presenting mean performance metrics with standard deviations is essential. As a suggestions, please complement these with confidence intervals and visualizations such as box plots or violin plots to illustrate the distribution of performance metrics across runs.
5) Please report all critical hyperparameters for all models and include a detailed description of the hyperparameter tuning methodology (if it was not performed please justify the strategy).
6) PLease use k-fold cross-validation in the hyperparameter tuning, ro ensure model generalizability.
7) Provide a table summarizing all hyperparameters and their final values for each model.
8) Include a summary table with statistical test results, including p-values or confidence intervals for model comparisons.
